# Ultra-inert lanthanide chelates as mass tags for multiplexed bioanalysis

Tomáš David [1], Miroslava Šedinová [1], Aneta Myšková [1,2], Jaroslav Kuneš [1,3], Lenka Maletínská [1], Radek Pohl [1], Martin Dračínský [1], Helena Mertlíková-Kaiserová[1], Karel Čížek[1], Blanka Klepetářová[1], Miroslava Litecká [4], Antonín Kaňa [2], David Sýkora [2], Adam Jaroš [1], Michal Straka[1] & Miloslav Polasek [1] ✉

Coordination compounds of lanthanides are indispensable in biomedical applications as MRI contrast agents and radiotherapeutics. However, since the introduction of the chelator DOTA four decades ago, there has been only limited progress on improving their thermodynamic stability and kinetic inertness, which are essential for safe in vivo use. Here, we present ClickZip, an innovative synthetic strategy employing a coordination-templated formation of a *1,5*-triazole bridge that improves kinetic inertness up to a million-fold relative to DOTA, expanding utility of lanthanide chelates beyond traditional uses. Acting as unique mass tags, the ClickZip chelates can be released from (biological) samples by acidic hydrolysis, chromatographically distinguished from interfering lanthanide species, and sensitively detected by mass spectrometry. Lanthanides enclosed in ClickZip chelates are chemically almost indistinguishable, providing a more versatile alternative to chemically identical isotopic labels for multiplexed analysis. The bioanalytical potential is demonstrated on tagged cell-penetrating peptides in vitro, and anti-obesity prolactin-releasing peptides in vivo.

Molecular labels have become an indispensable tool for interrogation of live matter. Unique labels have been developed to detect, quantify, or visualize molecules of interest with high sensitivity and specificity, including fluorescent tags[1,2], radioactive labels[3–5], nucleic acid-based labels[6,7], and mass tags[8–10]. Multiplexing—the concurrent use of multiple, distinguishable tags—opens up the possibility of more comprehensive analyses and deeper insight.

Multiplexing has been perfected in the controlled settings of in vitro studies (where the introduction of a molecular tag is generally inconsequential) and offers considerable potential for preclinical in vivo testing as well[4]. However, molecular recognition systems within a living organism can differentiate between closely related tags, thus influencing the in vivo behaviour of the studied labelled molecules and distorting the observations. Ideal molecular tags would appear

equivalent to the organism's molecular machinery, despite their individual uniqueness. Isotopic labelling meets this criterion, alas with significant constraints on multiplexing[8].

Lanthanides present a viable alternative to isotopic labels for molecular tagging. This group of fifteen elements, complemented by yttrium, exhibit exceptionally uniform chemistry. Aside from rare extremophile microorganisms[11], lanthanides play no biological role and maintain a low natural background ($\mu g\ L^{-1}$) in animal and human tissue[12–14]. They are recognized by living systems as a group, not as individual elements. Nonetheless, due to diverse physical properties, selected lanthanides find medical applications as MRI contrast agents[15] and radiotherapeutics[5], and their unique physico-chemical characteristics (such as luminescence[16,17], paramagnetism[18–20], and atomic mass[9,10]) make lanthanides promising for multiplexing.

[1]Institute of Organic Chemistry and Biochemistry, Czech Academy of Sciences, Prague, Czech Republic. [2]University of Chemistry and Technology Prague, Prague, Czech Republic. [3]Institute of Physiology, Czech Academy of Sciences, Prague, Czech Republic. [4]Institute of Inorganic Chemistry, Czech Academy of Sciences, Husinec-Řež, Czech Republic. ✉e-mail: miloslav.polasek@uochb.cas.cz

Utilizing lanthanides in vivo necessitates their encapsulation in chelates that are thermodynamically stable and kinetically inert[5,15]. The latter is more important for preventing metal dissociation in the dynamic microenvironment of living systems. The macrocyclic DOTA is the most widely used chelator satisfying these requirements. The primary concern is proton-assisted dechelation[21,22], which occurs at slow but significant rate at physiological pH, comparable to that of peptide bond hydrolysis[23,24]. This necessitates the development of more inert chelates, as highlighted by the quest for safer gadolinium-based MRI contrast agents[15] following the discovery of gadolinium retention in the brain[25]. Significantly increasing the inertness of these chelates could facilitate new applications that involve chemical conditions harsher than in vivo. For example, quantification of lanthanides in animal tissue requires acidic digestion of the biological matrix. DOTA chelates that release their metal in such conditions blur the distinction between the tag and background metal, complicating trace analysis due to the variable natural lanthanide content in tissues[12–14]. Unchelated metal detection then demands specialized techniques like ICP-MS or ICP-OES[9]. In contrast, more inert chelates, remaining intact post-digestion, would stand out from the background and be amenable to detection using widely available conventional mass spectrometers.

Attaining very high kinetic inertness in lanthanide chelates presents significant challenges, with only a few chelators to date surpassing the inertness of lanthanide DOTA chelates (Supplementary Fig. 1)[26–29]. This is typically accomplished using macrocyclic ligands rigidified by cross-bridging[28,30] or substituents that lock specific conformations[27]. However, these systems often present obstacles that hinder their use, such as multiple coordination isomers differing in properties[27,31]. There is also one fundamental challenge: while highly inert chelates possess a high activation barrier for metal dissociation,

this barrier also applies to chelate formation. Consequently, such chelates become increasingly difficult to make, requiring elevated temperatures and extended complexation times[27,28]. Fullerenes represent an ideal inescapable cage for metal ions, but such constructs cannot be obtained via direct complexation, which restricts their practical applicability[32].

In this work we introduce ClickZip, an innovative and scalable synthetic strategy providing ultra-inert lanthanide chelates, which employs a spontaneous, irreversible cage-locking reaction templated by the chelated metal ion. Their isostructural character and compatibility with a range of chemical reactions enable new applications in sensitive quantitative multiplexed bioanalysis, a concept we have validated both in vitro and in vivo.

## Results and discussion
### The ClickZip principle
Fast complexation and extremely high inertness are desirable but contradictory attributes to be achieved with conventional chelators. To circumvent this problem, we devised ClickZip as an unconventional molecular mechanism to irreversibly trap Ln[III] ions inside a coordination cage (Fig. 1A). A macrocyclic chelator was equipped with azide and alkyne moieties strategically placed on two opposing pyridine pendant arms. Although uncatalysed azide-alkyne cycloaddition is thermodynamically favoured, it is typically slow and requires rather forcing conditions[33]. The chelator thus remains open to readily accept Ln[III] ions for complexation. Upon complexation, the coordination of the pyridines sterically forces the azide-alkyne reaction, forming an intramolecular 1,5-triazole bridge locking the coordination cage. This reaction is notably different from copper(I)-catalysed click (the metal is not a catalyst−see further) and from the classical Huisgen cycloaddition (1,5-triazole is formed, compared to 1,4-/1,5-isomer mixture)[33], and can be

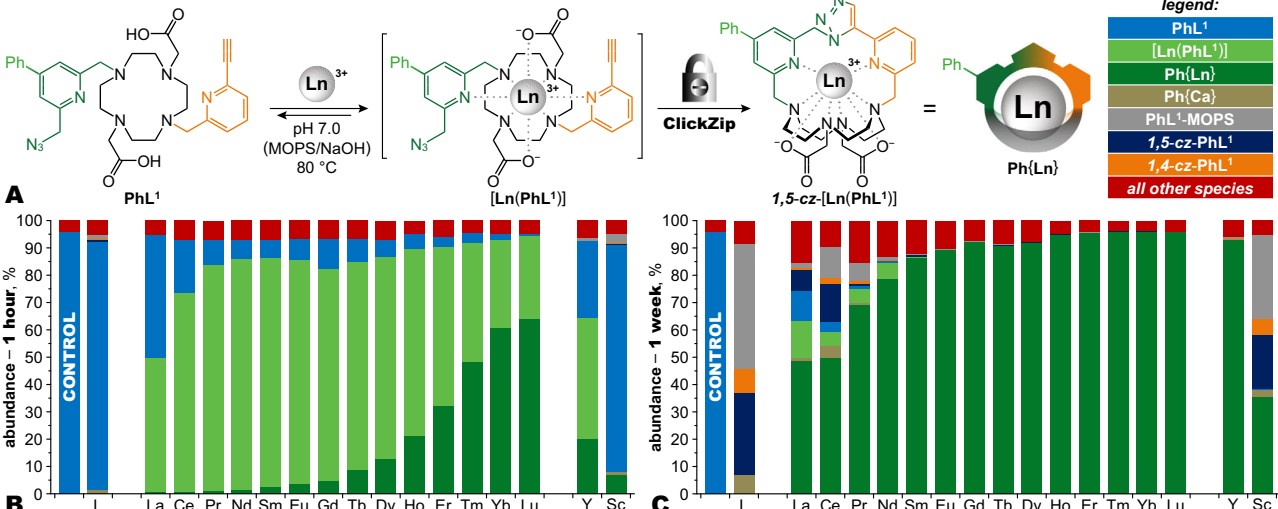

**Fig. 1 | The ClickZip principle and metal ion preferences. A** Chelator **PhL¹** acts as an irreversible molecular trap for Ln[III] ions in two steps: (1) binding the Ln[III] ion in a conventional and reversible complexation step; and (2) trapping the Ln[III] ion by irreversible formation of an intramolecular 1,5-triazole bridge that cross-links the two pyridine pendant arms, resulting in a cryptate-type chelate, **Ph{Ln}**. The whole process runs as one-pot procedure without isolation of the **[Ln(PhL¹)]** intermediate. **B**, **C** Percentage of different species formed during the ClickZip process as a function of the metal and time. **Conditions:** 0.5 mM **PhL¹** and 1.0 mM Ln[III] salt (including Y[III] and Sc[III]) in 50 mM aq. MOPS/NaOH buffer (pH 7.0) at 80 °C (except for column L, where no metal was added). **Analysis:** HPLC with UV detection at λ = 280 nm (further details in Supplementary Fig. 3). Starting purity of **PhL¹** prior to the experiment is labelled as CONTROL (details about **PhL¹** stability in Supplementary Fig. 4). Identified species are colour-coded as shown in the legend, with all other detected species jointly shown in red. **B** Results after 1 h at 80 °C. The efficacy

of ClickZip increases from largest La[III] ion to smallest Lu[III] ions. Note that the fastest **Ph{Lu}** is already formed with 65% conversion, while very little conversion to any product is observed for sample without any Ln[III] (column L). The corresponding intermediates **[Ln(PhL¹)]** are susceptible to acid hydrolysis during HPLC analysis (0.1% FA additive used in the mobile phase, pH -2.8) and therefore some fraction of **[Ln(PhL¹)]** is detected as **PhL¹**. **C** Results after 1 week at 80 °C. High conversion is achieved even for the slower-reacting Ln[III] ions (≥ 85% from Sm[III] to Lu[III] and Y[III]), with low amounts of side-products (red bar). The efficacy of ClickZip is low for metal ions that are too large (La[III] to Nd[III]) or too small (Sc[III]), yet the corresponding **Ph{Ln}** can still be isolated (except for very labile **Ph{Sc}**). The reaction without any Ln[III] (column L) provides a mixture of products. Identified side-products include **PhL¹-MOPS** adduct (see Supplementary Fig. 4), Ca[II] chelate **Ph{Ca}**, and empty cages (**1,4-cz-PhL¹** and **1,5-cz-PhL¹**). Source data available in Supplementary Data 2.

best described as a metal-templated Huisgen cycloaddition. Practically speaking, the ClickZip process runs as one-pot reaction in fully aqueous solution under heating (80 °C).

## Note on abbreviated notation

For brevity, the following notation will be used throughout the text for specific chemical species (see Supplementary Fig. 2): **L**[1] (open chelator); **RL**[1] (**L**[1] derivatized with R); [**M**(**RL**[1])] (open chelate of metal M with **RL**[1]); *1,4-cz*-**RL**[1] and *1,5-cz*-**RL**[1] (empty cages with *1,4-* and *1,5-* triazole bridge, respectively); *1,4-cz*-[**M**(**RL**[1])] and *1,5-cz*-[**M**(**RL**[1])] (ClickZip chelates with *1,4-* and *1,5-*triazole bridge, respectively). Unless important for isomer distinction, the most discussed *1,5-cz*-[**M**(**RL**[1])] is further abbreviated to **R**{**M**}.

## Metal ion role and preferences

Several observations point towards a templating rather than catalytic role of the Ln[III] ion in the ClickZip reaction. Firstly, the ClickZip rates and yields strongly depend on ionic radius, increasing from La[III] to Lu[III] (Fig. 1B-C, numerical values for all plots are available in Supplementary Data 2), with Y[III] falling between Dy[III] and Ho[III], confirming this trend. Secondly, using excess metal expedites the complexation, but the unchelated metal does not promote intermolecular triazole crosslinking. Thirdly, the purity, yield, and reaction rate of ClickZip are remarkably independent of concentration (5 μM–50 mM), with deviations notable only at the extreme limits (Supplementary Fig. 5). Overall, these results indicate that the chelated metal ion exerts indirect steric effects through coordination of the pyridines rather than participating directly in the azide-alkyne cycloaddition. It is worth noting that, for lanthanides from the end of the series, the efficacy and purity of the ClickZip reaction is easily amenable to upscaling (Supplementary Fig. 6).

## *1,4-/1,5-*Triazole selectivity

The surprising regioselectivity of ClickZip towards *1,5-*triazole products regardless of the Ln[III] choice and ligand derivatizations (Supplementary Fig. 3) prompted investigation with computational chemistry methods. The largest La[III], smallest Lu[III], and selected non-lanthanides (Ca[II], Li[I], Na[I], K[I]) were compared in terms of the calculated Gibbs free energies of their open intermediate [**M**(**L**[1])] and bridged *1,5-cz*-[**M**(**L**[1])] and *1,4-cz*-[**M**(**L**[1])] products. For all these metals there was a clear thermodynamic drive to both products, as expected for Huisgen cycloaddition. The *1,5-*isomer was favoured in all cases, except for the large La[III] and K[I] ions (Supplementary Figs. 7 and 8). However, experimental results proved that even La[III] provided exclusively the *1,5-*isomer (Fig. 1). This discrepancy was explained by considering the reaction mechanism and kinetics. The transition state leading to the *1,5-*triazole product was significantly lower in energy and therefore kinetically preferred in both La[III] and Lu[III], in agreement with the experimental results. The reason seems to be partial de-coordination of the pyridines required for both transition states, which is more pronounced and energetically demanding for the *1,4-*isomer (Supplementary Figs. 7 and 8, Supplementary Data 3). The peculiar case of alkaline metals will be discussed next.

## Empty cage synthesis and conventional complexation

To compare the ClickZip synthesis with conventional complexation, it was necessary to prepare the empty cage *1,5-cz*-**PhL1**. Interestingly, in the absence of Ln[III] ions, the intramolecular azide-alkyne reaction of **PhL**[1] yielded a mixture of species dependent on pH and buffer (Supplementary Fig. 9). The *1,5-cz*-**PhL**[1] was the major product under acidic pH or alkaline pH in presence of Na[I] ions. In the absence of Na[I] (using K[I], Rb[I] or Cs[I] in the buffer), alkaline pH resulted in dominant formation of the *1,4-cz*-**PhL**[1] isomer not observed with Ln[III] ions. In contrast, Li[I] ions suppressed formation of both products, largely preserving the open ligand **PhL**[1]. Thus, both empty-cage isomers could be obtained in high

yields under specific optimized conditions, confirmed by X-ray analysis (Supplementary Fig. 10). These results indicate that complexation of a size-matched metal ion (here Na[I]) or a specific degree of protonation (intramolecular hydrogen bonds) have similar templating effects towards *1,5-*triazole formation, while the absence of metal templating may provide the other isomer.

Direct complexation of Ln[III] ions with *1,5-cz*-**PhL**[1] was unsuccessful, with no **Ph**{**Ln**} product detectable after heating to 80 °C for 1 week or 6 months (Supplementary Figs. 11 and 12). Instead, formation of **Ph**{**Ca**} was observed, detected previously in trace amounts in reactions with Ln[III] ions (Fig. 1) and synthesis of empty cages (Supplementary Fig. 9), likely due to Ca[II] ions leaching from the glassware. This is in stark contrast to the *1,4-cz*-**PhL**[1] isomer, which provided *1,4-cz*-[**Ln**(**PhL**[1])] chelates by direct complexation of Ln[III] ions under the same conditions, though with mediocre yields. Lanthanide chelates of both types thus could be accessed via different strategies (Supplementary Fig. 13).

## Kinetic inertness

The kinetic inertness of the chelates was tested by acid-assisted dechelation under pseudo-first-order conditions with excess HCl, quantitatively monitored by LC-MS and expressed as half-lives. Chelates of a DOTA derivative [**Ln**(**NO₂BnDOTA**)], amenable to LC-MS detection, served as a reference[34]. Four increasingly demanding conditions from 0.1 M HCl at 25 °C to 6.0 M HCl at 80 °C were used to cover a broad range of half-lives. This revealed an increase in inertness across the series of **Ph**{**Ln**} chelates, spanning 10 orders of magnitude from La to Lu (Fig. 2A). Starting from Sm, **Ph**{**Ln**} surpassed inertness of the DOTA system, steadily improving up to Lu (Supplementary Figs. 14–18). **Ph**{**Lu**} showed very high resistance to dechelation even under the harshest conditions (Fig. 2B). With an estimated half-life of 3 years in 6.0 M HCl at 80 °C, this system exhibits greater inertness than other lanthanide chelates previously reported as highly inert[27–29]. In contrast, the isomeric *1,4-cz*-[**Ln**(**PhL**[1])] chelates were much less kinetically inert, approximately 2 orders of magnitude worse than the DOTA system. Inertness of selected **Ph**{**Ln**} chelates (Ln = Eu, Ho, Lu) was also tested under transmetallation conditions with Zn[II] and Cu[II] ions (Supplementary Fig. 19). No reaction with Zn[II] and only a few percent of Cu[II] chelate were observed after 1 week at 80 °C with 10-fold excess of these metals, further confirming high inertness of the chelates even towards this mechanism of dechelation.

## Solid-state structures, isostructurality, isomerism

The striking differences in properties between the *1,5-*triazole- and *1,4-* triazole-bridged chelates are best understood from their solid-state structures (Fig. 3). In the case of **Ph**{**Lu**}, the *1,5-*triazole is part of an 18-membered ring, where all five donor N-atoms (three from cyclen, two from pyridines) can coordinate tightly to the Lu[III] ion. On the other hand, the *1,4-*triazole in *1,4-cz*-[**Lu**(**PhL**[1])] chelate increases the size of this ring to 19 atoms, bringing steric strain and chain conformations that disfavour simultaneous coordination of both pyridines. This mismatch explains why the *1,4-cz*-[**Ln**(**PhL**[1])] chelates are not produced via the ClickZip reaction and are much less inert. The orientation of the triazole hydrogen relative to the coordination cage may also play a role (Fig. 3).

Solid-state structures for all **Ph**{**Ln**} chelates from Sm to Lu (including Y) revealed notable similarity of the molecules (Supplementary Fig. 20), despite the dramatic differences in kinetic inertness. Structural parameters showed very small relative changes in response to the lanthanide contraction; the coordination environments and overall shapes of the molecules remained the same. This isostructurality was corroborated by the behaviour of the chelates in reversed-phase chromatography, where a mixed sample of **Ph**{**Ln**} chelates (Ln = Sm to Lu, Y) showed a single peak with no sign of

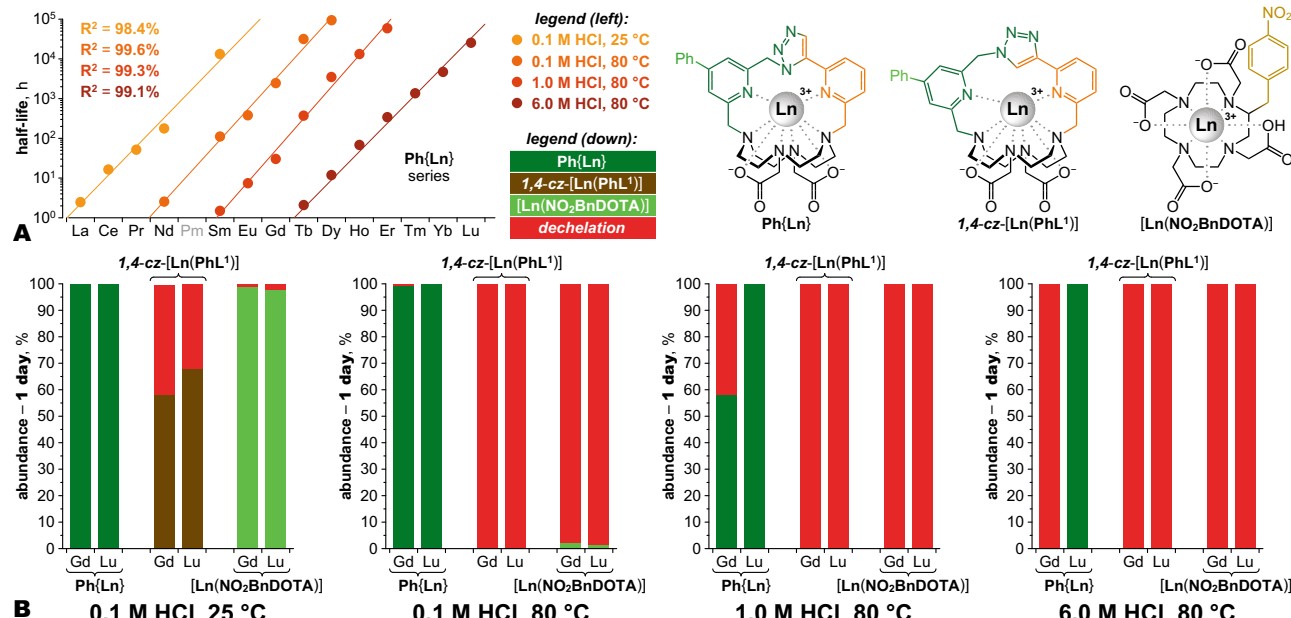

**Fig. 2 | Kinetic inertness.** Comparison of **Ph{Ln}** series and other Ln[III] chelates in terms of acid-assisted dechelation. *Conditions:* 0.5 mM chelates under four different conditions (from 0.1 M HCl at 25 °C to 6.0 M HCl at 80 °C). *Analysis:* HPLC with UV detection at $\lambda$ = 278 nm for **Ph{Ln}** and *1,4-cz*-[**Ln(PhL¹)**] series, and $\lambda$ = 285 nm for sum of two diastereomers of [**Ln(NO₂BnDOTA)**] (further details in Supplementary Fig. 14). The [**Ln(NO₂BnDOTA)**] system serves as a reasonable surrogate for [**Ln(DOTA)**], which is not amenable to the same analysis for lack of a UV chromophore and retention on reversed-phase HPLC. **A** Kinetic inertness of **Ph{Ln}** expressed as dechelation half-life for given conditions. On the logarithmic scale, the values show a linear growth within the lanthanide series (radioactive Pm[III]

not tested), allowing extrapolation of ~10 orders of magnitude difference between the least inert **Ph{La}** and most inert **Ph{Lu}** chelates (numeric values in Supplementary Fig. 14). **B** Degree of de-chelation (expressed as % of intact metal chelate under given conditions) for **Ph{Ln}**, *1,4-cz*-[**Ln(PhL¹)**] and [**Ln(NO₂BnDOTA)**] systems. Only Gd[III] and Lu[III] chelates are shown for simplicity (full data set with more species in Supplementary Figs. 15–18). The **Ph{Ln}** chelates, and particularly **Ph{Lu}**, demonstrate inertness vastly surpassing [**Ln(NO₂BnDOTA)**]. In contrast, the *1,4-cz*-[**Lu(PhL¹)**] chelates are less inert than [**Ln(NO₂BnDOTA)**]. Source data available in Supplementary Data 2.

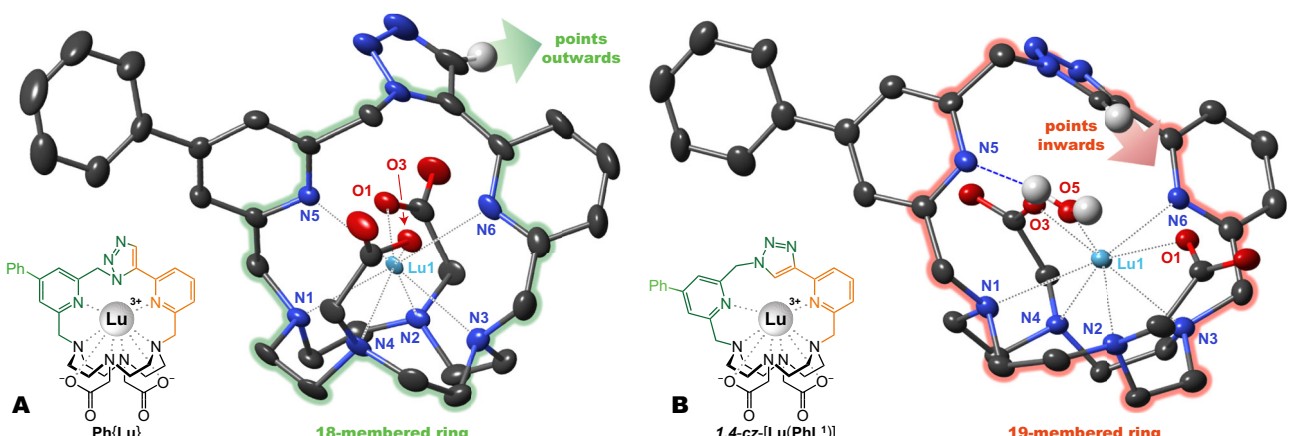

**Fig. 3 | Solid-state structures of ClickZip chelate isomers.** Comparison of crystal structures of **Ph{Lu}** and *1,4-cz*-[**Lu(PhL¹)**] highlighting dominating features of the Lu[III] ion coordination environment. Hydrogen atoms (except for triazole moiety and water ligand) and labels of non-coordinating atoms were omitted for clarity. Thermal ellipsoids were set at 50% probability. Both compounds crystallized in centrosymmetric groups containing two enantiomeric chelate units with opposite chirality of the coordination cage in TSA conformations $\Lambda\lambda\lambda\lambda\lambda$ and $\Delta\delta\delta\delta\delta$, of which only one is shown. **A** Structure of [**Ph{Lu}**]⁺ cation ($\Delta\delta\delta\delta\delta$ isomer) found in the crystal structure of [**Ph{Lu}**]⁺[ClO₄]⁻·7.0H₂O. The *1,5*-triazole is part of an 18-membered macrocyclic ring that allows coordination of both pyridines. The hydrogen atom of the triazole points away from the coordination cage. Due to the

snug fit around the Lu[III] ion, coordination of water is not allowed. **B** Structure of *1,4-cz*-[**Lu(PhL¹)**](H₂O)⁺ cation ($\Lambda\lambda\lambda\lambda\lambda$ isomer) found in the crystal structure of [*1,4-cz*-[**Lu(PhL¹)**](H₂O)]⁺[ClO₄]⁻·2H₂O·*i*-PrOH. The *1,4*-triazole is part of an irregularly-shaped 19-membered macrocyclic ring that disfavours simultaneous coordination of both pyridines. Due to a poor fit of the pyridine-triazole-pyridine bridge to the metal, one water molecule is coordinated to the Lu[III] ion, forming a hydrogen bond (dashed blue line) to the de-coordinated pyridine. The hydrogen atom of the triazole is turned towards the coordination cage, possibly interfering with the coordination sphere. Note that this solid-state structure may not represent the situation in solution, where equilibrium may exist between coordination of water or pyridine.

separation (Supplementary Fig. 21). Only the early lanthanides from La to Nd demonstrated notable deviations.

The ClickZip chelates are inherently chiral, with defined rotation of the pendant arms (Λ or Δ) and of the four ethylene units in the cyclen ring (λλλλ or δδδδ), analogous to the DOTA system[15,27,34]. However, in contrast to DOTA, they only adopt enantiomeric ΛΛΛΛΛ and Δδδδδ forms with twisted-square antiprismatic (TSA) arrangements (Fig. 4). To probe whether these enantiomers can interconvert, we modified {**Lu**} with L-cysteine residues to produce a pair of distinguishable diastereomers. These were chromatographically separated and their epimerization was followed by NMR, revealing that they interconvert within hours at 37 °C (Supplementary Fig. 22), much slower than the DOTA chelates[31]. The enantiomers of {**M**} can thus be regarded either as one or two compounds, depending on the time scale.

## Post ClickZip synthesis

Their high stability and synthetic accessibility make ClickZip chelates interesting substrates for further chemical transformations. A range of reactions could be performed on the organic ligand while keeping the Ln[III] ion safely locked inside (Fig. 5). Heating **Ph**{**Lu**} with excess strong base (DBU) in $D_2O$ led to full deuteration of all five $CH_2$ groups in the pendant arms by proton exchange. Extensive treatment with $NaBH_4$ in MeOH quantitatively reduced one of the coordinated pyridines to a piperidine ring, which remained coordinated to the chelated metal. Functional groups exposed to the exterior could undergo efficient transformations including Suzuki coupling, nucleophilic aromatic substitutions, and copper(I)-catalysed click, showcasing the versatility of ClickZip chelates as synthetic building blocks.

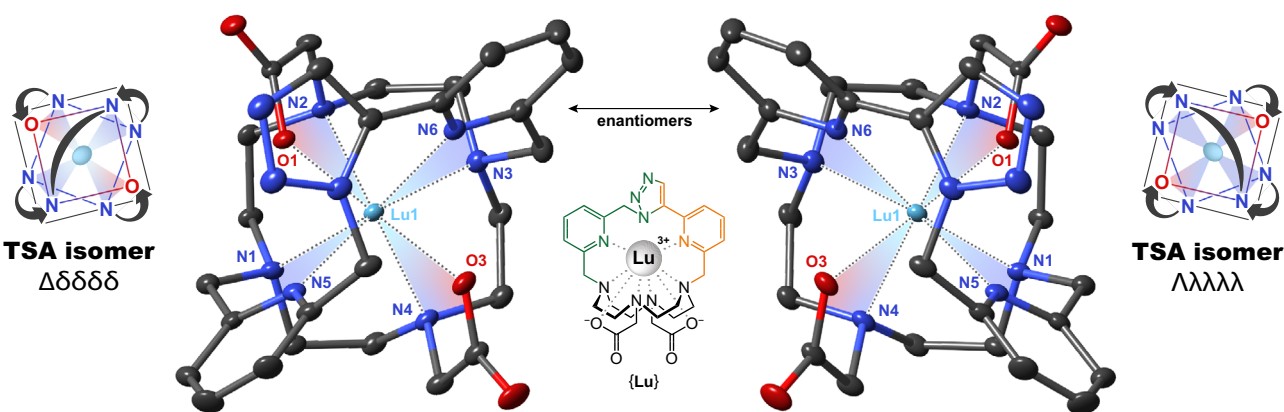

**Fig. 4 | Chirality of ClickZip chelates.** Inherent chirality of the molecules is given by rotation of the pendant arms (Λ or Δ) and of the four ethylene units in the cyclen ring (λλλλ or δδδδ). In theory, four isomers are possible: two twisted square antiprismatic (TSA) enantiomers (ΛΛΛΛΛ and Δδδδδ), and two square antiprismatic (SA) enantiomers (Δλλλλ and Λδδδδ). The ClickZip chelates exclusively adopt the TSA arrangements ΛΛΛΛΛ and Δδδδδ, which were observed together in all solid-state structures as well as in solution. These two forms interconvert on the time scale of hours at 37 °C (Supplementary Fig. 22). Both enantiomers shown here were found in the crystal structure of [{**Lu**}]⁺[ClO₄]⁻·5H₂O. Thermal ellipsoids were set at 50% probability.

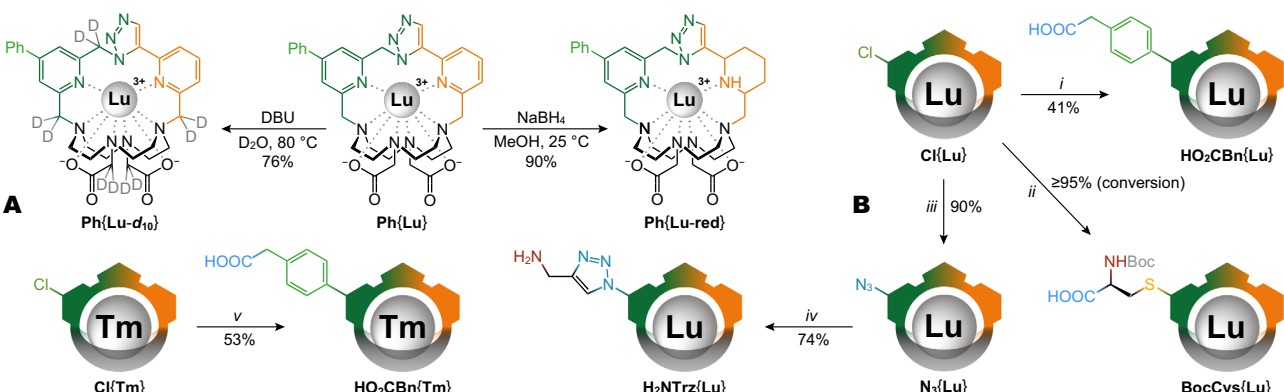

**Fig. 5 | Post ClickZip synthesis.** Demonstration of the high robustness of ClickZip chelates with possible derivatization of its core or surface. No leak of Ln[III] ion was observed during any of these transformations. Yields refer to isolated compounds (except for the case of **BocCys**{**Lu**} where conversion is given instead). **A** Examples of core modifications (full structures). In addition to their unusually high stability in strong acids, ClickZip chelates can withstand other harsh conditions, even with its {**Ln**} core undergoes chemical transformation. Prolonged heating of **Ph**{**Lu**} with the excess (>30 equiv.) of strong base DBU (*1,8*-diazabicyclo[5.4.0]undec-*7*-ene) in $D_2O$ allowed exchange of all acidic carbon-bound hydrogen atoms for deuterium (>95% H to D exchange after single evaporation/$D_2O$ resupply cycle) without loss of the Lu[III] ions. Similarly, when a large excess of strong reducing agent (>1000 equiv. of $NaBH_4$) was applied to **Ph**{**Lu**} in MeOH, quantitative reduction of one of

the pyridine rings was achieved, yet no Lu[III] ions were found to escape from the cage during the process. **B** Examples of surface modifications (symbol structures). The ClickZip core is so benign that its presence can be ignored when performing surface modifications. For example, the **Cl**{**Ln**} was found to be a versatile precursor for the introduction of a wide range of functional groups (COOH, NH₂, N₃ and amino-acid) onto the ClickZip chelate, though it showed unexpected dechlorination in presence of formate, which should be avoided (Supplementary Fig. 23). **Conditions:** (*i*) 4-(carboxymethyl)phenylboronic acid pinacol ester, XPhos Pd G2 (cat.), K₃PO₄, DMF, H₂O, 80 °C; (*ii*) *N*-Boc-cysteine, DIPEA, DMSO, RT; (*iii*) NaN₃, DMSO, 80 °C; (*iv*) propargyl amine, CuSO₄ (cat.), sodium ascorbate, MES/NaOH buffer (pH = 5.2), H₂O, RT; (*v*) 2-(4-boronophenyl)acetic acid, XPhos Pd G2 (cat.), K₃PO₄, DMF, H₂O, 80 °C.

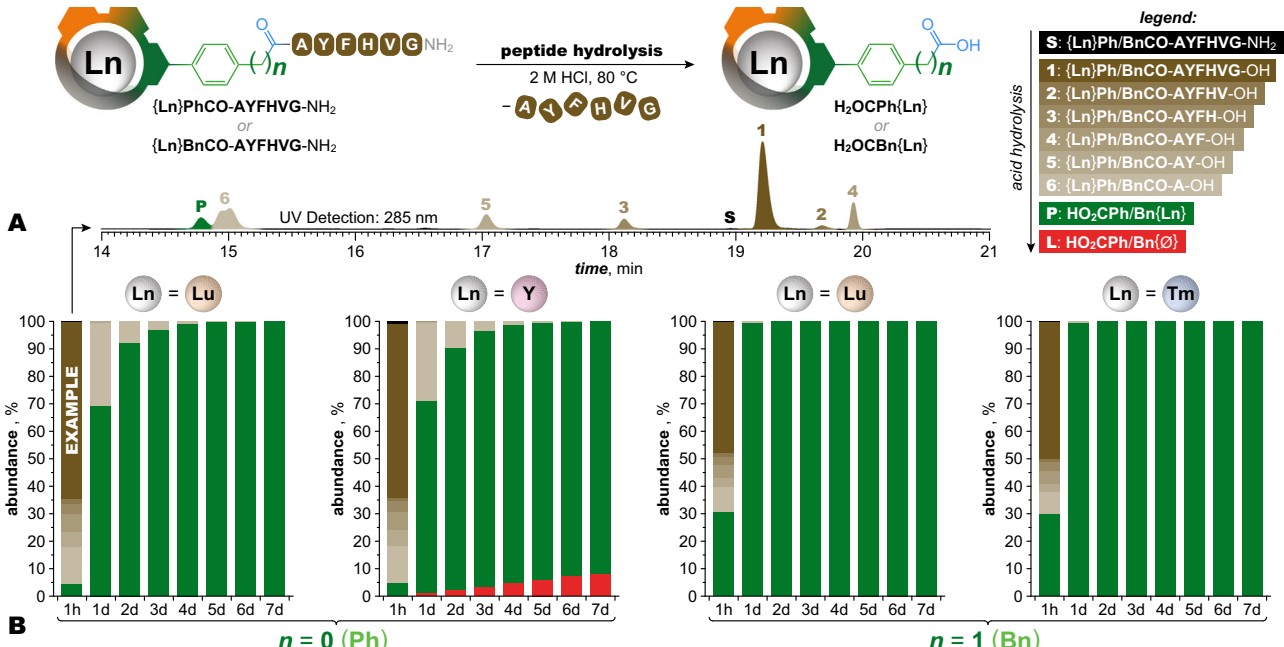

**Fig. 6 | ClickZip and peptide bond stability comparison.** Hydrolysis of ClickZip conjugates with model hexapeptide **AYFHVG**-NH$_2$ connected by either aromatic or aliphatic amide bonds in strong acid. *Conditions:* 0.5 mM {**Ln**}**PhCO**-**AYFHVG**-NH$_2$ or {**Ln**}**BnCO**-**AYFHVG**-NH$_2$ in 2 M HCl at 80 °C. *Analysis:* $\lambda$ = 285 nm (HPLC, H$_2$O–MeCN, formic acid additive). **A** Sample HPLC chromatogram of ongoing hydrolysis of {**Lu**}**PhCO**-**AYFHVG**-NH$_2$ (after 1 h in 2 M HCl at 80 °C) with all ClickZip-containing species distinguished (fragments containing only amino acids do not interfere with the peak integration as their UV absorption at 285 nm is negligible). **B** Excerpt from hydrolysis of model ClickZip hexapeptide conjugates recorded over the course of 7 d shows that complete liberation of intact ClickZip (with Tm$^{III}$ or Lu$^{III}$ ion) from its peptide conjugate is possible (the stability of ClickZip surpasses that of the peptide bond). The hydrolysis is faster for {**Ln**}**BnCO**-

**AYFHVG**-NH$_2$ species connected to the peptide via aliphatic amide moiety ($n = 1$) where **HO$_2$CBn**{**Ln**} is formed quantitatively after 1 d. For the aromatic analogues ($n = 0$), the complete liberation proceeded more slowly, with the hydrolysis of the last amino-acid from {**Ln**}**PhCO**-**A**-OH intermediate being the rate-limiting step. Nevertheless, prolonged reaction time led to quantitative formation of **HO$_2$CPh**{**Ln**}. Although a small leak of Y$^{III}$ ions accompanied the hydrolytic process of {**Y**}**PhCO**-**AYFHVG**-NH$_2$ ( ~ 5% after 4 d when the hydrolysis of the last amino acid was nearly completed), nearly intact Y$^{III}$ chelate could be anticipated if coupled with the use of aliphatic amide moiety ($n = 1$) instead. Additional hydrolytic conditions can be found in Supplementary Fig. 24. Source data available in Supplementary Data 2.

## Compatibility with peptide hydrolysis

While the acid-assisted dechelation of lanthanide DOTA chelates proceeds with a rate similar to hydrolysis of a peptide bond[24], the ultra-inert ClickZip chelates of Lu$^{III}$ and near lanthanides should outlast complete hydrolysis of peptides and proteins. To test this, we prepared model hexapeptide conjugates with two ClickZip derivatives, **HO$_2$CPh**{**Ln**} or **HO2CBn**{**Ln**}, which differed in their amide linkage to the peptide *N*-end (Fig. 6). Hydrolysis in 2 M HCl at 80 °C digested the peptide to individual amino acids; the chelates were cleaved from the peptide with no loss of the metal (Lu$^{III}$ or Tm$^{III}$), or only small loss (5% for Y$^{III}$ chelate). Faster cleavage of **HO$_2$CBn**{**Ln**} shows that the tags can be optimized for this step.

## Multiplexing in vitro

For their resistance to acidic hydrolysis and isostructural character, ClickZip chelates are ideal tags for multiplexed bioanalysis. We selected cell internalization as a biological effect to test whether tags carrying different metals are distinguishable to living cells (Fig. 7A). Two cell-penetrating peptides (CPPs), based on hexaarginine and its partially protected derivative, were labelled with {Lu} and {Tm} tags at the *N*-termini, resulting in a total of four conjugates (Fig. 7B). Paired comparisons for compounds carrying different tags were performed simultaneously on the same sample of cells (Fig. 7C). Tags released from acidic digestion of the cells were quantified by LC-MS (Fig. 7D). As expected, the peptide with partially protected guanidine groups was internalized less efficiently than the naked hexaarginine. However, compounds with identical peptides but different tags were internalized into cells to a similar extent, regardless of their cell-penetrating

efficiency. When compounds with different peptides were compared, permutating the tags provided a mirror image of the results. This demonstrates that the observed differences in the biological effect (internalization) were due to differences in the peptide part and the choice of metal had a negligible effect. The LC-MS quantification was verified by ICP-OES (Fig. 7E).

## Ex vivo quantification in animal tissue

Compared to in vitro cell cultures, quantification of mass tags in animal tissue must deal with much more complex matrices of interferences. To demonstrate that ClickZip chelates are compatible with such conditions, we used two 31-amino acid prolactin-releasing peptide (PrRP31) analogues previously studied for their anti-obesity effect[35], which differ in the presence of a biodistribution-altering fatty acid residue: native PrRP31 or PrRP31 palmitoylated at position 11 (palm$^{11}$-PrRP31). The two peptides were labelled with derivatized {Lu} and {Tm} tags (Fig. 8A) and intravenously (*i.v.*) administered to C57BL/6 J mice independently or in a mixture (Fig. 8B). Liver tissue was selected for ex vivo analysis, as it has the highest natural lanthanide background of all organs, presenting a particularly difficult matrix[12]. Although pre-purification of the tags was possible, we opted for analysis of the full lysate, removing only insolubles (Supplementary Figs. 26 and 27). Both tags were confidently quantified on an LC-MS/MS system primarily used for proteomics (Fig. 8C, method validation in Supplementary Data 1). However, independent verification with elemental analysis on ICP-MS provided a different view. The levels of Lu and Tm were higher than expected, even in samples to which they were not applied (Fig. 8D, method validation in Supplementary Data 1). Here, the ability

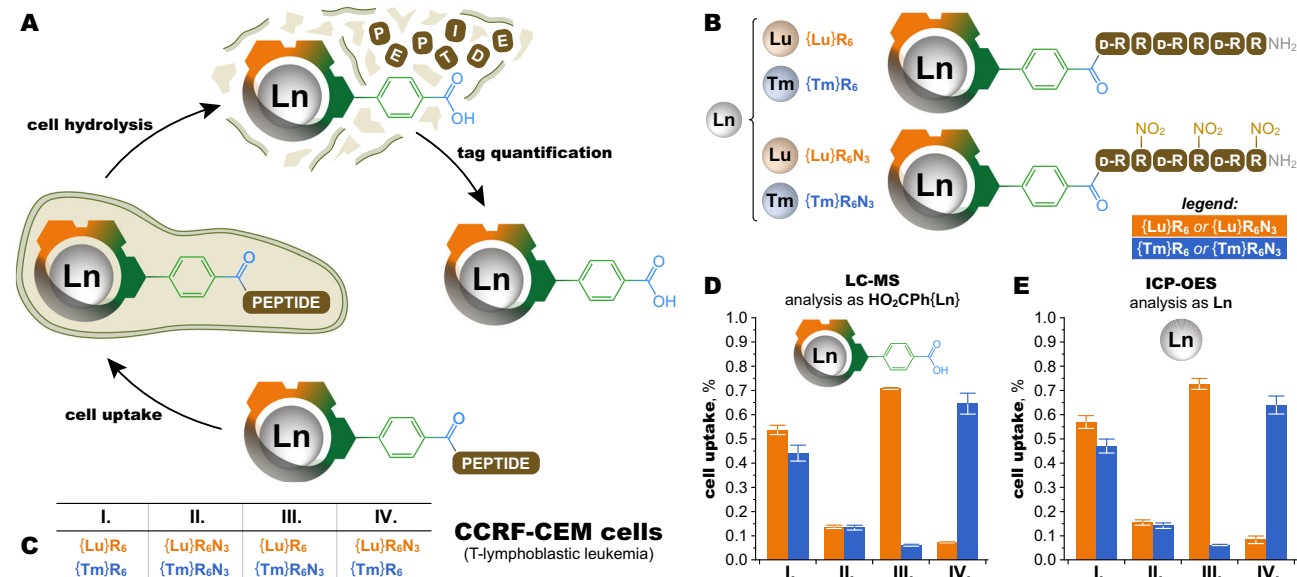

**Fig. 7 | ClickZip-tagged peptide use and quantification in cell cultures.**
**A** Schematic depicting the use of ClickZip chelates as mass tags for quantitative analysis of labelled peptides. The key aspect is that the tag is quantitatively released into solution from both the peptide and the cells by total acidic hydrolysis and quantified in the form of an intact metal Ln[III] chelate. **B** Four tagged model peptide conjugates were prepared based on two cell-penetrating hexapeptides (different sequences of L-arginine, D-arginine and NO₂-protected L-arginine) labelled with Lu- or Tm-containing ClickZip tags. **C** Pairs of conjugates were internalized to CCRF-CEM cells (in triplicates), followed by total acidic hydrolysis (as shown in (**A**)) before tag quantification. **D** LC-MS (liquid chromatography−mass spectrometry, single quadrupole) quantification of the intact ClickZip tags in cell

lysate demonstrates on paired comparisons that internalization of the conjugates is a function of the peptide sequence, not of the metal in the ClickZip tags. Additional details on the method of analysis are in Supplementary Fig. 25. **E** The same samples from (**D**) were analysed with ICP-OES (inductively coupled plasma−optical emission spectroscopy) for direct detection of Lu and Tm, showing excellent agreement with results from panel D and demonstrating that quantification of ClickZip tags with commonplace LC-MS instrumentation provides results equal to more specialized and less common techniques. For (**D**) and (**E**), the data are presented as the mean ± SD ($N = 3$, measurements on independent cell lysates from $10^6$ cells each). Source data available in Supplementary Data 2.

to distinguish lanthanides in different chemical forms proved essential. Using liquid chromatography prior to ICP-MS detection, the {Ln} tags could be discerned from the unchelated Ln[III] ions, confirming the same {Ln} content as determined by LC-MS/MS (Fig. 8E, Supplementary Fig. 31). Because other lanthanides were found in the liver samples only at much lower levels, it was suspected that the unchelated Lu and Tm were contaminants, rather than biological background. The exact source could not be traced, but likely originated from environmental contamination in a laboratory that regularly works with lanthanides. In conventional trace-metal analysis, contamination may completely obscure the desired measurement. However, the {Ln} tags demonstrate advantageous analytical robustness, allowing quantitative separation and subtraction of the unchelated lanthanides, regardless of their origin.

### In vivo multiplexing capabilities
To probe the degree of similarity between ClickZip tags under in vivo conditions, we labelled the lipidized peptide palm[11]-PrRP31 with {Tm}, {Yb}, or {Lu} tags, and followed them with ICP-MS for 60 minutes after *i.v.* co-administration to Wistar rats (Fig. 9A, B). The pharmacokinetics of all three conjugates in blood plasma were nearly identical, with the highest concentration observed immediately after injection, followed by rapid clearance[36] (Fig. 9C). The three lanthanides were further determined in two liver lobes (caudate and left lateral) after the animals were sacrificed. The same amounts were found in both lobes and the relative ratios of the metals were not significantly different (Fig. 9D). Thus, varying the Ln[III] ion in ClickZip tags has a negligible effect on both pharmacokinetics and biodistribution of tagged biomolecules, allowing reliable tracking and quantification of multiple labelled molecules simultaneously in vivo.

ClickZip is a principle for irreversible entrapment of lanthanide ions, producing coordination compounds with stability up to million-

fold greater than in current lanthanide-based MRI contrast agents and radiotherapeutics. The ClickZip chelates effectively encapsulate the Ln[III] ions in an organic shell, enabling them to be manipulated as typical organic compounds in both synthetic and analytical procedures. As ultra-inert molecular tags, ClickZip chelates can be differentiated from interfering lanthanide natural background and contamination, qualifying for ultra-trace analysis in biological and other open systems. ClickZip may substantially expand the scope of use of lanthanide chelates in organic synthesis, diagnostic, and therapeutic agents, and other areas requiring the highest possible stability and inertness.

## Methods
### Liquid chromatography
Quantification of ClickZip tags was performed on Exion LC AD with tandem mass triple quadrupole mass spectrometer (QTrap 6500 + ) from *Sciex* (referred to as LC-MS/MS) equipped with Kinetex column (1.7 μm F5, 100 × 2.1 mm) and a guard column (1.7 μm F5, 2.1 mm; both from *Phenomenex*) using 5 mM aq. AF with 0.1% FA−5 mM AF with 0.1% FA in MeOH gradient (0.2 mL min⁻¹ flow rate). Analytes were monitored through their specific multiple reaction monitoring (MRM) transitions by utilizing electrospray ionization (ESI) in a positive mode (Supplementary Fig. 28). Quantification of both **HO₂CPh{Lu}** and **HO2CPh{Tm}** ClickZip tags was conducted using a weighted calibration curve ($1/x^2$) with an internal standard **HO₂CPh{Y}**.

### Inductively coupled plasma mass spectrometry
ICP-MS experiments were performed on NexION 350D instrument from *PerkinElmer*. The sample introduction system consisted of a peristaltic pump, concentric nebulizer and a cyclonic spray chamber. Syngistix™ 1.1 software from *Perkin Elmer was* used for the data acquisition and data evaluation.

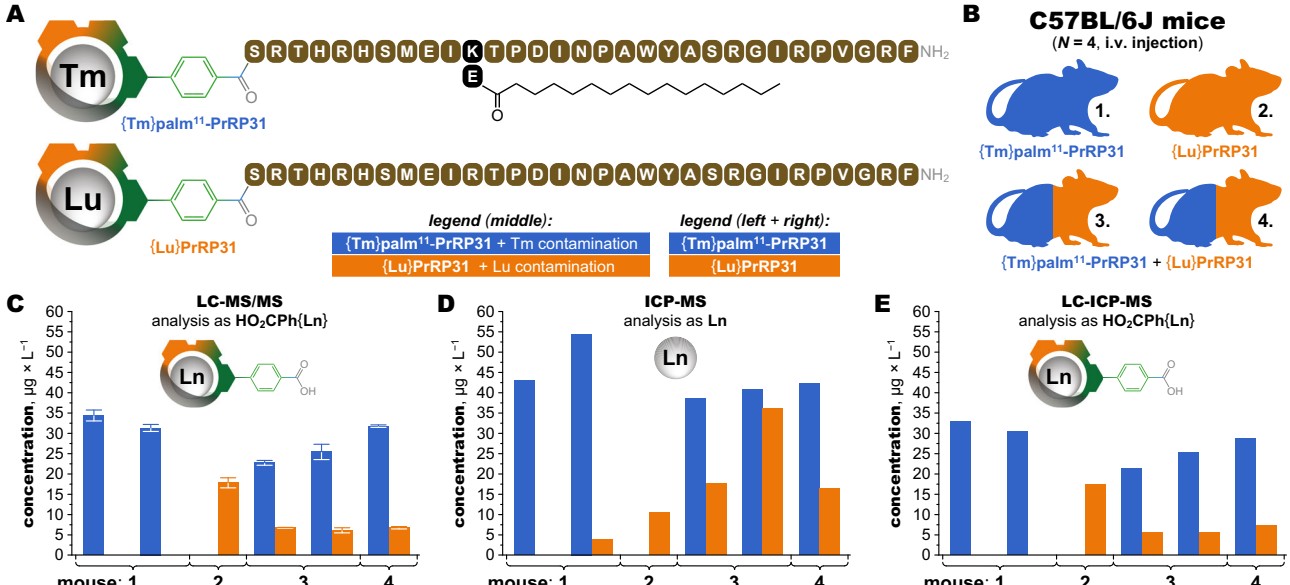

**Fig. 8 | Ex vivo analysis of tagged peptides in animal tissue. A** Two anti-obesity drug candidates based on 31-amino-acid prolactin-releasing peptide were labelled with Lu- and Tm-containing ClickZip tags, providing two conjugates: **{Tm}palm[11]-PrRP31** and **{Lu}PrRP31**. **B** Injection scheme for four C57BL/6 J mice that received either a single conjugate or mixture of both simultaneously (*i.v.* injection, each compound at 500 µg/kg dose). Animals were sacrificed 5 min post-injection, samples of liver harvested and hydrolysed in HCl as detailed in Methods, and the tags quantified in the lysate. **C** Quantification of ClickZip tags with a proteomics-level LC-MS/MS (liquid chromatography−tandem mass spectrometry). For mice 1 and 3, two liver samples were taken and are shown independently to show repeatability within the same animal and tissue type. Each analysis was repeated three times, the data are presented as the mean ± SD (*N* = 3, technical replicates). In all cases, the tags were confidently quantified in accordance with the injection scheme and expected higher liver uptake of the lipidized conjugate (details, including LOD/LOQ values, in Supplementary Figs. 28 and 29). **D** Quantification with ICP-MS (inductively coupled plasma−mass spectrometry), which sensitively detects the chemical elements (details, including LOD/LOQ values, in Supplementary Fig. 30), showed higher concentrations compared to results from panel A, indicating potential contamination with Lu and Tm in chemical forms other than the ClickZip tags. **E** The use of liquid chromatography (LC) prior to ICP-MS analysis confirmed speciation of the metals into the ClickZip tags and contaminating unchelated Ln[III] ions. Chromatographic separation of the ClickZip tags allowed their quantification independently of the free Ln[III], achieving excellent agreement with LC-MS/MS analysis from (**C**) (Supplementary Fig. 31). The concentrations in (**C**)−(**E**) represent µg of Tm or Lu found in liver lysate (100 mg of liver in total volume of 400 µL, see Methods for details). Source data available in Supplementary Data 2.

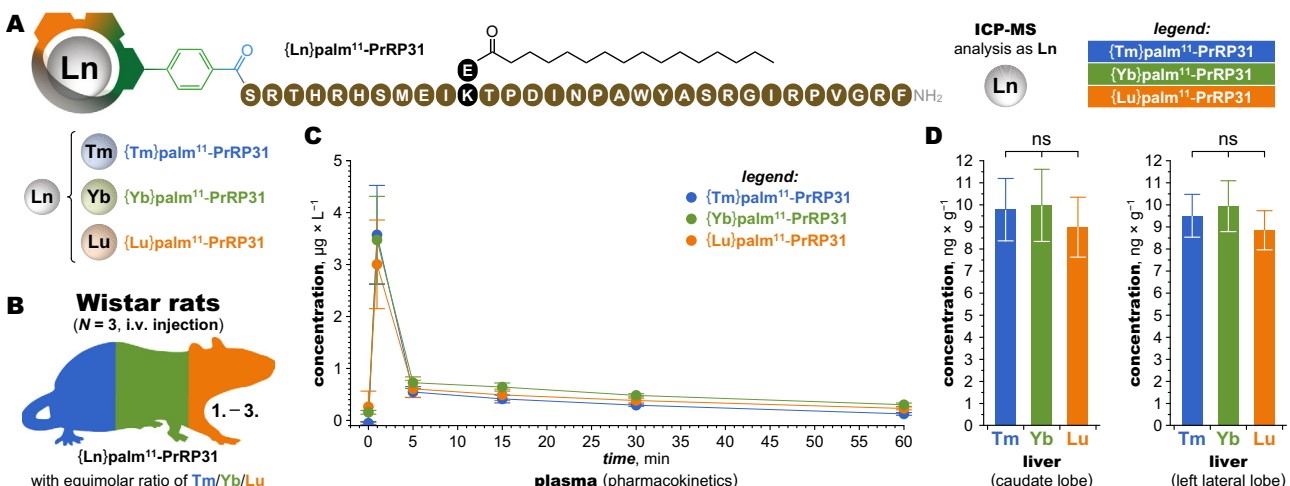

**Fig. 9 | Pharmacokinetics and liver uptake of {Ln}-palm[11]-PrRP31 in Wistar rats analysed by ICP-MS. A** Lipidized peptide palm[11]-PrRP31 was labelled with ClickZip tags to provide three **{Ln}palm[11]-PrRP31** conjugates differing only in the metal (Ln = Tm, Yb, Lu). **B** Wistar rats (*N* = 3) were injected *i.v.* with a mixture of the three conjugates (each at a dose of 0.03 mg/kg). **C** Pharmacokinetics in blood plasma with collection at 1, 5, 15, 30 and 60 min post-injection analysed by ICP-MS (inductively coupled plasma−mass spectrometry) shows identical fast clearance for the three conjugates regardless of the metal in the {Ln} tag. **D** Biodistribution in two liver lobes (caudate and left lateral) at 60 min post-injection are nearly identical, demonstrating homogenous distribution of the tag quantification within this organ and excellent reproducibility of the tag quantification. The data are presented as the mean ± SD. One-way ANOVA followed by Bonferroni´s multiple comparisons test confirmed that the content of the tags is not significantly different (ns), showing that tags containing different metals behave comparably in vivo. Source data available in Supplementary Data 2.

## Liquid chromatography−inductively coupled plasma mass spectrometry

LC-ICP-MS analysis was performed using high-pressure pump Series 200 from *Perkin-Elmer*, a degasser, a sampling valve from *IDEX Health & Science LLC*, equipped with PEEK sample loop and a chromatographic column Luna® Omega Polar C18 (5 μm, 100 Å, 150 × 4.6 mm) by *Waters*. Isocratic elution using 5% *1,2*-hexanediol in $H_2O$ with 1% FA as a mobile phase (1 mL min$^{-1}$ flow rate).

## Synthesis

Detailed synthetic procedures, including characterization of all products, are provided in Supplementary Figs. 32–124. The experimental section for each compound is accompanied by $^1H$ NMR spectrum (HPLC chromatogram given instead for paramagnetic chelates or not isolated compounds) and UV absorption profile from analytical HPLC (with the exception of intermediates). All syntheses proceeded in solution except for solid phase peptide synthesis of H-[D-R(Pbf) R(Pbf)]₃-NH₂, and peptides PrRP31 and palm$^{II}$-PrRP31 (previously reported)[37].

## pH dependent formation of empty cages

Stock solution (5 mM in $H_2O$) of freshly prepared ligand **PhL$^1$** was mixed in a glass vial (2 mL) with buffer (pH range: 2.2–7.0 citrate/ NaOH; 8.6–10.2 borate/NaOH). Mixture was diluted with $H_2O$ to reach final concentration of 0.5 mM ligand in 50 mM buffer (overall volume 500 μL). For pH 0 and 1, a final concentration of 0.5 mM ligand in 1.0 M and 0.1 M $HClO_4$ was used (HCl was avoided due to a possible reaction with alkyne). Resulting solutions were capped and stirred at 80 °C. Reaction mixtures were analysed by HPLC (C18; $H_2O$–MeCN gradient with FA additive) after 1 hour and 1 day. Influence of alkali metals was studied analogously at pH 10.2 with borate/$M_2CO_3$ buffer system (M = Li–Cs). Aliquots of reactions mixtures were diluted with $H_2O$, briefly vortexed and directly used for the analysis.

## Lanthanide dependent formation of ClickZip chelates

Stock solution (5 mM in $H_2O$) of freshly prepared ligand (**L$^1$**, **ClL$^1$**, **HO₂CPhL$^1$** or **PhL$^1$**) was mixed in a glass vial (2 mL) with MOPS/NaOH buffer (pH 7.0) followed by addition of $LnCl_3$ solution (100 mM). Mixture was diluted with $H_2O$ to reach final concentration of 0.5 mM ligand and 1.0 mM Ln$^{III}$ (including Y$^{III}$ and Sc$^{III}$) in 50 mM buffer (overall volume 500 μL). An additional vial with ligand and buffer (but without any metal ion) was included in each series. Resulting solutions were capped and stirred at 80 °C. Reaction mixtures were analysed by HPLC (C18; $H_2O$–MeCN gradient with FA additive) after 1 hour and 1 week. Aliquots of reactions mixtures were diluted with $H_2O$, briefly vortexed and centrifuged before the analysis.

## Direct lanthanide complexation with empty cages

Stock solution (5 mM in $H_2O$) of empty cage (**1,4-cz-PhL$^1$** or **1,5-cz-PhL$^1$**) was mixed in a glass vial (2 mL) with MOPS/NaOH buffer (pH 7.0) followed by addition of $LnCl_3$ solution (100 mM). Mixture was diluted with $H_2O$ to reach final concentration of 0.5 mM ligand and 1.0 mM Ln$^{III}$ (including Y$^{III}$ and Sc$^{III}$) in 50 mM buffer (overall volume 500 μL). An additional vial with Ca$^{II}$ instead of Ln$^{III}$ was included in both series. Resulting solutions were capped and stirred at 80 °C. Reaction mixtures were analysed by HPLC (C18; $H_2O$–MeCN gradient with FA additive) after 1 hour and 1 week (and up to 6 months for **1,5-cz-PhL$^1$** with Gd$^{III}$ and Lu$^{III}$). Aliquots of reactions mixtures were diluted with $H_2O$, briefly vortexed and centrifuged before the analysis.

## Kinetic inertness

Stock solution (0.5–5.0 mM in $H_2O$) of isolated analyte (with the exception of [**Ln(NO₂BnDOTA)**] solutions that were prepared in situ) was mixed in a glass vial (2 mL) with HCl (titrated stock solution or its derived stock solution) and $H_2O$ to reach final concentration of 0.5 mM

analyte in either 0.1 M, 1 M or 6 M HCl (overall volume 500 μL). In case of 6 M HCl, the stock solution of analyte in the vial was evaporated to dryness on high vacuum before the addition of the acid. Resulting set of solutions were sealed with a stopper with a septum and stirred at either 25 °C (0.1 M HCl) or 80 °C (0.1 M, 1.0 M and 6.0 M HCl). Reaction mixtures were analysed by HPLC (C18; $H_2O$–MeCN gradient with FA additive) after 1 hour, 1 day, 1 week and 1 month (30 days). The aliquots of reactions mixtures from **Ph{Ln}** (including **Ph{Y}**) were neutralized (as no re-complexation could occur) with MOPS/NaOH buffer (pH 7.0), briefly vortexed and centrifuged before the analysis. To prevent re-complexation of other species (and thus false negative results), the aliquots were diluted with $H_2O$ (for reaction mixtures in 0.1 M HCl) or partly neutralized by FA/NaOH buffer (pH 3.6, for reaction mixtures in 1.0 M or 6.0 M HCl) before the analysis (this was verified by repeated analysis of the same diluted sample which showed identical species ratios). Obtained experimental data were fitted by exponential decay function $y = \exp(-x/t)$ using Origin 9 ($y$ = fraction of intact chelate, $x$ = time, $t$ = decay constant). Kinetic inertness was then expressed as half-life $t_{1/2}$ (in hours) derived from decay constant ($t_{1/2} = \ln(2) \times t$).

## Hydrolysis of ClickZip conjugates with model hexapeptides

Stock solution (1 mM in $H_2O$) of given model hexapeptide conjugate was mixed in a glass vial (2 mL) with HCl (titrated stock solution) and $H_2O$ to reach final concentration of 0.5 mM (**{Ln}PhCO-AYFHVG-NH₂** or **{Ln}BnCO-AYFHVG-NH₂** in either 1 M, 2 M or 3 M HCl (overall volume 240 μL). Resulting set of solutions were sealed with a stopper with a septum and stirred at 80 °C. Reaction mixtures were analysed by LC-MS (C18; $H_2O$–MeCN gradient with FA additive) after 1 hour and then after each day for 1 week. The aliquots of reaction mixtures were neutralized (as no re-complexation could occur) with MOPS/NaOH buffer (pH 7.0) and briefly vortexed before the analysis.

## Cell experiments

CCRF-CEM cells (CCL-119TM) purchased from ATCC (Manassas, VA, USA) are human T lymphoblasts isolated from the peripheral blood of a 4-year-old Caucasian female with acute lymphoblastic leukemia (ALL). The cell line was authenticated by the manufacturer. Cross-contamination in the laboratory was prevented by renewing of the cells from early passage cryovials every 3 months. The cells were negatively tested for Mycoplasma (MycoAlertTM Mycoplasma detection kit, Lonza). Grown cells were centrifuged (250 × g, RT), washed with PBS and resuspended in RPMI medium Dutch modification (with HEPES buffer, without antibiotics) to a concentration $2.2 \times 10^7$ mL$^{-1}$. Suspension was transferred into 12 (for 4 combinations of CPP conjugates, each in triplicate) Eppendorf tubes (450 μL of suspension to each). Pairs of given CPP conjugates in equimolar amounts were used in each combination: **{Lu}R$_6$** + **{Tm}R6** (combination I); **{Lu} R₆N₃** + **{Tm}R₆N₃** (combination II); **{Lu}R₆** and **{Tm}R₆N₃** (combination III); **{Lu}R₆N₃** + **{Tm}R₆** (combination IV). Stock solutions of corresponding CPP conjugates (5.0 mM in RPMI medium, 50 μL) were added to the cell suspension to reach final concentration 0.5 mM given of CPP conjugate (overall volume 500 μL). Cells were then incubated for 2 h at 37 °C. Following the incubation, compound uptake was terminated by centrifuging the samples (900 × g, RT). Medium was carefully pipetted out, the pellets were washed twice with 1 mL PBS and then treated with HCl (3.0 M, 500 μL) for hydrolysis. Resulting set of solutions were stirred at 80 °C for 1 week. The aliquots of reactions mixtures were evaporated and re-dissolved in MOPS/NaOH buffer (pH 7.0). The resulting clear solutions were (after vortexing) analysed by LC-MS (C18; $H_2O$–MeCN gradient with FA additive). The total amounts of CPP conjugates incorporated into cells was then indirectly quantified by sum of UV peak area of **H₂OCPh{Lu}** and **H₂OCPh{Tm}** whose retention time is identical (concentrations were obtained from calibration curve of standards of known concentration). The ratio of

**H₂OCPh{Lu} / H₂OCPh{Tm}** was then obtained from ratio of EIC⁺ peak area of both analytes in MS. The cell penetrating properties of hexaarginine peptides were then expressed as a ratio between found amounts of **H₂OCPh{Ln}** and starting amounts of given CPP conjugate.

## Animal experiments

All animals were obtained from Charles River (Sulzfeld, Germany). The animal experiments followed the ethical guidelines for animal experiments in the Czech Republic Act Nr. 246/1992 and were approved by the Committee for Experiments with Laboratory Animals of Czech Academy of Sciences under the protocol number 80/2020 (experiment name: Relationship between obesity, diabetes and neurodegeneration: new therapeutic potential of prolactin releasing peptide analogues). Animals were housed under standard laboratory conditions (temperature $23 \pm 1\,°C$, 12 h light/dark cycle). They were fed with a standard laboratory chow (R-M-H diet, Sniff, Germany) and had a free access to tap water. Sex was not considered a variable expected to influence outcomes, as the experiments were focused on demonstrating the performance of analytical methods rather than investigating biological effects. Statistical analysis was performed in *GraphPad Prism* (version 8.4.3).

Male C57BL/6 J mice at age of 6 months ($N = 4$) were anesthetized with pentobarbital and subsequently injected *i.v.* into a jugular vein with a **{Tm}palmᴵᴵ-PrRP31**, **{Lu}PrRP31** or a mixture of **{Tm}palmᴵᴵ-PrRP31** and **{Lu}PrRP31** all dissolved in saline at 0.5 mg mL⁻¹. Then 15 min post-injection blood was collected by cardiac puncture, perfusion with saline supplemented with heparin (10 U/mL, Zentiva, Prague, Czech Republic) was carried out and a dissection was performed. Mouse liver (~100 mg) were transferred into 6 glass vials (4 mL) followed by addition of H₂O and HCl (titrated stock solution) to reach suspension of the liver in 6 M HCl (overall volume 1.0 mL). Resulting set of suspensions were sealed with a stopper with a septum and stirred at 80 °C for 3 days. Reaction mixtures were then carefully evaporated on rotary evaporator to remove most of HCl. The residues were suspended in MOPS/NaOH buffer (250 mM; pH 7.0; 2 mL) to neutralize the remaining acid and the resulting suspension were vigorously vortexed. Resulting mixtures were then filtered into new sets of vials (20 mL) using syringe microfilter (RC) and the residues and filters were further washed with H₂O (3 × 1 mL). Resulting new sets of vials (with ~5 mL volume in each) were concentrated to ~1 mL followed by transfer into a new set of vials (4 mL). The residues were further washed with H₂O (2 × 0.5 mL). Resulting new sets of vials (with ~2 mL volume in each) were carefully evaporated to dryness and the residues were re-dissolved in H₂O (400 μL). Resulting set of solutions were then analysed for **HO₂CPh{Tm}** and **HO₂CPh{Lu}** content: aliquot from each sample (4.0 μL) was mixed with **HO₂CPh{Y}** internal standard (2.0 μL of 1.0 μM stock solution to compensate the matrix effect of liver lysate) and with NH₄OAc (14.0 μL of 10 mM stock solution). The mixture was briefly vortexed and 10 μL of the resulting solution was used for LC-MS/MS analysis. Values in Fig. 8 are reported as μg × L⁻¹ of elemental Tm or Lu in liver lysate, corrected for variations in actual liver sample weights, which ranged from 76.4 to 112.2 mg, to correspond to exactly 100 mg of liver in 400 μL of lysate.

Male Wistar rats at age of 3 months ($N = 3$, 300-350 g) were used for the pharmacokinetic experiment; details are described in literature[36]. Day before experiment, polyethylene catheter (PE 10) was inserted to the left jugular vein for *i.v.* administration of mixture **{Ln}palmᴵᴵ-PrRP31** and PE 50 catheter was inserted to the left a. carotid for blood collection. Both catheters were exteriorized in the interscapular region. Blank blood plasma was obtained from all animals just before **{Ln}palmᴵᴵ-PrRP31** application. The design of the pharmacokinetic experiment was as follows: 3 rats were injected *i.v.* with a mixture of three differently labelled palmᴵᴵ-PrRP31, specifically

**{Tm}palmᴵᴵ-PrRP31**, **{Yb}palmᴵᴵ-PrRP31** and **{Lu}palmᴵᴵ-PrRP31**, each at a dose of ~0.03 mg kg⁻¹, dissolved in saline. The *i.v.* administration was performed into cannulated jugular vein. Then at 1, 5, 15, 30 and 60 minutes post-injection, blood was collected from cannulated carotid artery. All the blood samples were collected to Eppendorf tubes with pre-cooled EDTA and centrifuged (10,000 g, 5 min and 4 °C) to prepare blood plasma. At 60 min rats were sacrificed by decapitation and two liver lobes (caudate and left lateral) were collected. All samples were stored at −80 °C prior to analysis. A sample of liver lobe (0.5 g) was digested with 3 mL of 65% nitric acid in a Teflon® vessel in a Speedwave 4 microwave digestion system from *Berghof* for 10 min at 190 °C. The dissolved samples were supplemented with Y(NO₃)₃ as an internal standard and diluted to 25 mL with deionized water. All the plasma samples from pharmacokinetics were diluted 21× and spiked with Y(NO₃)₃ as an internal standard.

## Reporting summary

Further information on research design is available in the Nature Portfolio Reporting Summary linked to this article.

## Data availability

The crystallographic data for the structures reported in this study have been deposited at the Cambridge Crystallographic Data Centre (CCDC), under the deposition numbers 2334554 (**1,5-cz-PhL¹**), 2334555 (**1,4-cz-PhL¹**), 2334547 (**{Lu}**), 2334558 (**Ph{Y}**), 2334561 (**Ph{Lu}**), 2334552 (**Ph{Yb}**), 2334562 (**Ph{Tm}**), 2334556 (**Ph{Er}**), 2334559 (**Ph{Ho}**), 2334560 (**Ph{Dy}**), 2334553 (**Ph{Tb}**), 2334563 (**Ph{Gd}**), 2334544 (**Ph{Eu}**), 2334546 (**Ph{Sm}**), 2334550 (**Ph{Pr}**), 2334557 (**Ph{Ca}**), 2334545 (**1,4-cz-[Lu(PhL¹)]**), 2334548 (**Cl{Lu}**), 2334549 (**HO₂CPh{Lu}**) and 2334551 (**HO₂CBn{Tm}**). These data can be obtained free of charge from the Centre via its website (www.ccdc.cam.ac.uk/getstructures). The validation data for analytical methods are provided in the Supplementary Data 1. All data used to create plots and charts are provided in numerical form in the Supplementary Data 2. Atomic coordinates of the optimized computational models are provided in Supplementary Data 3. Additional details for methods, synthesis and characterization of new compounds, and X-ray structures are provided in Supplementary Information. All other data supporting the findings in this study are available within the article and Supplementary Information. All data are available from the corresponding author upon request.

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

## Acknowledgements

This work was supported by the project National Institute for Research of Metabolic and Cardiovascular Diseases (Programme EXCELES, ID Project No. LX22NPO5104)—Funded by the European Union—NextGenerationEU (M.P. and L.M.), and RVO:61388963 and RVO:67985823 of Czech Academy of Sciences (institutional support). Czech Science Foundation Project No. 21-23261S is also acknowledged (M.S. and A.J.).

## Author contributions

M.P. conceived the project and supervised the work. T.D. synthesized the compounds, performed kinetic experiments, HPLC and LC-MS measurements and created the artwork/graphics. T.D. and M.P. wrote the manuscript. M.Š. assisted compound synthesis and analyses. M.D. and R.P. performed NMR experiments. K.Č. performed quantitative detection by LC-MS/MS. H.M.K. performed in vitro cellular uptake experiments. B.K. and M.L. performed the X-ray diffraction analysis. M.S. and A.J. performed DFT calculations and analysis. L.M., J.K. and A.M. performed in vivo experiments. A.M., A.K. and D.S. performed ICP-MS and LC-ICP-MS measurements. Authors thank Kelsea Grace Jones for language editing. All authors proofread and approved the manuscript.

## Competing interests

M.P., T.D., M.S. and A.J. are co-inventors on a patent application no. PCT/CZ2022/050087 filed in the name of applicant Ustav Organicke Chemie a Biochemie AV CR V.V.I. The application covers some of the compounds and their use discussed in this work. M.P., T.D., M.S. and A.J. declare no other competing interests. The other authors declare no competing interests.
