## [Transparent Peer Review file · Nature Communications]

Ultra-inert lanthanide chelates as mass tags for multiplexed bioanalysis

Corresponding Author: Dr Miloslav Polasek

Version 0:

Reviewer comments:

Reviewer #1

(Remarks to the Author)

This paper presents an innovative approach in which click chemistry is used to generate a macrobicyclic structure that traps lanthanide ions, providing complexes with exceptionally inertness. For the late lanthanide ions, the authors demonstrate an astonishing inertness, even in very harsh conditions (6 M HCl, 80 °C). The manuscript presents a huge amount of work, but it is easy to read, providing a more detailed discussion as supporting information. It also provides good and balanced citations. The work is technically sound and the concept is definitively new. I thus recommend publication of this paper in Nat Commun. I have just a few minor comments that the authors should address:

- 1) Page 3, bottom: The authors state: ClickZip process runs as one-pot reaction in fully aqueous solution under moderate heating (80 °C). I think 80 °C is a rather high temperature, and thus suggest to remove the word moderate. I actually believe that the use of milder conditions could be potentially very useful for different applications.
- 2) Computational details: The authors indicate that they used the MWB60 pseudopotential, and provide ref. 45. However, this pseudopotential was defined for the actinides (ref. 45). The authors probably used the large core pseudopotential that includes 60 electrons in the core for Lu, but 46 for La. Please correct.
- 3) Kinetic inertness: Lanthanide complexes with macrocyclic ligands generally dissociate following the acid-catalyzed mechanism. However, it would be useful to check whether other mechanisms operate here (i. e. metal-assisted in the presence of Cu(II)), at least in a qualitative way. The N atoms of the triazol ring could facilitate the formation of a ternary complex with Cu, providing a pathway for complex dissociation. I think this should be checked to confirm that these complexes are actually very inert.

Reviewer #2

(Remarks to the Author)

This is an innovative study describing the formation of exceptionally inert Ln complexes. The authors demonstrate potential utility in a number of areas. The work is extremely thorough and clearly presented. I congratulate the authors on an exceptional coordination chemistry study.

I have two minor comments:

1. I was surprised by the lack of complex formation with Sc(III). Accepting that this is a smaller ion, I also wonder if the chelation conditions were appropriate for this extremely acidic ion which will be insoluble at the pH used. Could the authors comment on whether they tried conditions at lower pH or in the presence of weak Sc(III) chelators. In other words, is the inefficient complexation of Sc(III) a fundamental property of the ligand/complex or is it a result of the reaction conditions?
2. Can the authors comment on whether they see a possibility for Ln ion separation based on the kinetic differences shown in Fig 1?

Reviewer #3

(Remarks to the Author)

The authors report an innovative approach to highly inert lanthanoid chelates, based on a complexation-triggered alkyne-azide cycloaddition. The intriguing and highly selective cycloaddition leads to the formation of cryptate-like Ln(III)-chelates, with an almost irreversible encapsulation of the metal ion.

The manuscript describes a monumental work with a careful and systematic investigation of the so-called ClickZip reaction with respect to the efficiency through the whole lanthanoid series (expanded to Sc and Y), the robustness towards the ligand substitution pattern and the isolation and characterization of the byproducts. The inertness of the Click-Zipped chelates is extensively studied and proved. The impressive stability of these chelates is demonstrated by their resistance to post-ClickZip chemical transformations and to the acidic hydrolytic conditions used for the complete hydrolysis of peptides. The last property has been exploited by tagging peptides with Ln(III)-chelates for in vivo studies, later recovering and quantifying the intact chelates.

In my opinion the manuscript deserves publication in Nature Communications, with only the following minor revisions.

Comments and corrections

- 1) All the metal chelates (along with the chelating agents L1s and their immediate macrocyclic precursors) have been isolated by reverse-phase preparative HPLC, even if the ClickZip reaction is demonstrated to be highly selective and efficient. This is a strong limitation to the potential applications of these useful highly inert chelates. I am well aware that macrocyclic chelates are really difficult to isolate and purify, sometimes requiring dedicated techniques, that are nevertheless been scaled up and industrialized. The authors should report at least an example of a larger scale preparation (1 mmol or more) of one of the L1 chelating agents/metal chelates.
- 2) Suppl. Fig. 3, Effect of L1 substitution on the Ln(III) ClickZip reaction. The ClickZip reaction of CIL1 with the early lanthanoids appears to be plagued by significant amounts of byproducts, with a clear difference of the reaction pattern observed in the ClickZip reaction of other L1s. Have the author an explanation for this difference? Have the authors any additional indication on the byproducts of this specific reaction?
- 3) The triazole ring in the structures of the 1,5-cz-[Ln(L1)] systems is drawn with a high degree of distortion. Please follow the IUPAC recommendations (10.1351/pac200880020277) for graphical representation standards for chemical structure diagrams, in order to have an aesthetic depiction. The same recommendations should be applied to the drawing of the neo-formed double bond in PhL1-MOPS (Suppl. Fig. 1, Suppl. Fig. 4) and to its corresponding (undefined) stereochemistry.
- 4) Page S30 and following pages, synthetic procedures,
 - ¹H-NMR spectra. Please report coupling constants with one decimal digit.
 - ¹³C-NMR signals should be better reported with the corresponding multiplicity (C/CH/CH₂/CH₃, referred to the non-decoupled ¹³C spectrum) rather than generic terms as "arom" and "mc".
- 5) Introduction: "in-vivo", "in-vitro", please remove hyphen
- 6) Page 5: "proton(iz)ation"
- 7) Page S12, the triazol(e) signal
- 8) Page S30, desym(m)etrization
- 9) Page S30, please replace "TMS≡" with the explicit form "ethynyltrimethylsilane" used afterwards.
- 10) Please use a uniform symbol for the volume (ml/mL) through the manuscript and the supplementary information.

Version 1:

Reviewer comments:

Reviewer #1

(Remarks to the Author)

The authors have provided detailed responses to all comments included in my previous report. They corrected a few minor issues and performed additional experiments to analyze the metal-assisted dissociation pathway. The results show that Zn(II) does not trigger complex dissociation at all, while Cu(II) does induce minor dissociation in very harsh conditions. Overall, these new results confirm the exceptional inertness of the complexes claimed in this paper. The manuscript is now ready for publication in Nat Commun.

Reviewer #2

(Remarks to the Author)

Thank you for addressing my comments

Reviewer #3

(Remarks to the Author)

Comments to revisions.

1) Preparation/RP-HPLC/larger scale.

The authors acknowledge that HPLC purification may present a limitation for a potential scale-up of the synthesis; personally, I agree that it is not an unsurmountable one with such an efficient transformation and favourable impurity profiles. A larger-scale preparation is now reported, demonstrating as expected the scalability of the ClickZip reaction.

2) Effect of L1 substitution on the Ln(III) ClickZip reaction.

The authors provide a reasonable explanation of the higher percentage of byproducts observed in the ClickZip reaction of CIL1 with early lanthanoids.

3) Drawing, Triazole and MOPS-byproduct

I understand the difficulty, almost impossibility, to have a drawing of the whole molecule with the chelate moiety and the triazole both without distortions. I agree with the authors to give a priority to a correct drawing of the chelate substructure with a symmetrical arrangement of the pyridine rings, at the expense of the drawing of the triazole ring.

4) - ¹H-NMR spectra. Please report coupling constants with one decimal digit.

The authors reported the coupling constants with one decimal digit.

Please specify the meaning of the non-standard abbreviation “dm” for the signals at 7.94 ppm and 8.05 ppm of compounds 12,13.

5) ¹³C-NMR signals should be better reported with the corresponding multiplicity (C/CH/CH₂/CH₃) rather than generic terms as “arom” and “mc”.

The authors fixed most of the multiplicities. Some generic non-multiplicity terms (“Ph”) remain and should be corrected (see compounds: 6,7,8,9, PhL1, cz-1,4-PhL1, cz-1,5-PhL1, 1,4/1,4-cz-[Lu(PhL1)]) and the diamagnetic Ph{Y}, Ph{Lu}, Ph{La}, Ph{Ca}, Ph{Lu-red}).

6) All typos and other minor mistakes have been corrected.

In my opinion the manuscript deserves publication in Nature Communications, once the remaining minor corrections are applied.

Point-by-point response to the REVIEWER COMMENTS

for revision of the manuscript

"Ultra-inert lanthanide chelates as mass tags for multiplexed bioanalysis "

Reviewer #1 (Remarks to the Author):

This paper presents an innovative approach in which click chemistry is used to generate a macrobicyclic structure that traps lanthanide ions, providing complexes with exceptionally inertness. For the late lanthanide ions, the authors demonstrate an astonishing inertness, even in very harsh conditions (6 M HCl, 80 °C). The manuscript presents a huge amount of work, but it is easy to read, providing a more detailed discussion as supporting information. It also provides good and balanced citations. The work is technically sound and the concept is definitively new. I thus recommend publication of this paper in Nat Commun. I have just a few minor comments that the authors should address:

1) Page 3, bottom: The authors state: ClickZip process runs as one-pot reaction in fully aqueous solution under moderate heating (80 °C). I think 80 °C is a rather high temperature, and thus suggest to remove the word moderate. I actually believe that the use of milder conditions could be potentially very useful for different applications.

We agree that "moderate" is probably inappropriate for 80 °C; the word was deleted.

This temperature was selected as a compromise between reaction speed and convenience to work safely in closed vessels below the boiling point of water. In fact, the reaction proceeds even at lower temperatures, obviously with lower speed. So, depending on the particular conditions that the reviewer has in mind, it may still be possible to use ClickZip at milder conditions, especially with the lanthanides from the end of the series.

2) Computational details: The authors indicate that they used the MWB60 pseudopotential, and provide ref. 45. However, this pseudopotential was defined for the actinides (ref. 45). The authors probably used the large core pseudopotential that includes 60 electrons in the core for Lu, but 46 for La. Please correct.

We thank the reviewer for catching this mistake, which was corrected. This part of methods has now been moved to Supplementary Information/Supplementary Methods, including the said references, which are now in Supplementary Information/Supplementary References.

3) Kinetic inertness: Lanthanide complexes with macrocyclic ligands generally dissociate following the acid-catalyzed mechanism. However, it would be useful to check whether other mechanisms operate here (i. e. metal-assisted in the presence of Cu(II)), at least in a qualitative way. The N atoms of the triazol ring could facilitate the formation of a ternary complex with Cu, providing a pathway for complex dissociation. I think this should be checked to confirm that these complexes are actually very inert.

Based on this comment we conducted a new set of experiments that we detail in the Supplementary Fig. 19 and we added a short commentary to the "Kinetic inertness" section. We used quite forcing conditions: 10-fold excess of either Cu(II) or Zn(II) ions relative to the ClickZip chelate (Ln(III) = Eu, Ho, Lu), and 80 °C. After 1 week, there was no detectable dechelation or transmetallation caused by the Zn(II), so the ClickZip chelates are extremely inert towards this mechanism. For the reaction with Cu(II) ions, a few percent of the Cu(II) chelate was observed after 1 week. Under these very forcing conditions the Cu(II) ions indeed cause slow complex dissociation. However, the degree to which Cu(II) causes dechelation/transmetallation is still very small. Under *in vivo* conditions this mechanism likely has negligible effect.

We thank the reviewer for this useful suggestion, which allowed us to demonstrate the exceptional inertness of the ClickZip chelates from a different perspective.

Reviewer #2 (Remarks to the Author):

This is an innovative study describing the formation of exceptionally inert Ln complexes. The authors demonstrate potential utility in a number of areas. The work is extremely thorough and clearly presented. I congratulate the authors on an exceptional coordination chemistry study.

I have two minor comments:

1. I was surprised by the lack of complex formation with Sc(III). Accepting that this is a smaller ion, I also wonder if the chelation conditions were appropriate for this extremely acidic ion which will be insoluble at the pH used. Could the authors comment on whether they tried conditions at lower pH or in the presence of weak Sc(III) chelators. In other words, is the inefficient complexation of Sc(III) a fundamental property of the ligand/complex or is it a result of the reaction conditions?

Scandium(III) makes an odd case in the whole series because of its significantly smaller ionic radius, as the reviewer correctly assumes. We found that this metal ion has a templating effect and produces the corresponding ClickZip chelate under the same conditions as the other tested metals. However, the resulting chelate is so kinetically labile that it decomposes (dechelates) even under the mildly acidic conditions that we used for the HPLC analysis (0.1% formic acid in the mobile phase). The chelate is thus detected on HPLC partly as the intact chelate and partly as the empty *1,5-cz-PhL*¹ cage. The free cage is a result of dechelation during the analysis and the ratio between these two species varies depending on the HPLC method. Both effects, the templating and dechelation, can be seen in Figure 1 by comparing the mixture of products observed with Sc(III) to the ligand without metal (column "L"). It is worth noting that despite of its small size, the Sc(III) ions could not be chelated by direct complexation, as shown in Supplementary Fig. 11.

That said, we observed the ClickZip reaction with Sc(III) ions to be slightly "cloudy", indicating solubility issues with this acidic ion, as the reviewer suggests. Please also note that we used 2 equivalents of the metal relative to the ligand, so partial precipitation of the excess metal can be expected. To avoid this precipitation, we tested the reaction at lower pH = 5.2 (data not shown, will be part of a follow-up paper). These conditions indeed accelerated formation of the ClickZip chelate with Sc(III) ions compared to pH = 7.

Nevertheless, the poor inertness of the Sc(III) chelate remains a significant issue. The chelate could not even be isolated by preparative HPLC, which we used as a standard method for all other chelates. Thus, given its extreme lability and therefore poor prospect for any practical use, we decided not to study the Sc(III) case further and not optimize either conditions of its formation, nor conditions for its isolation.

The poor inertness of **Ph{Sc}** chelate is mentioned in the caption to Fig. 1: " The efficacy of ClickZip is low for metal ions that are too large (La^{III} to Nd^{III}) or too small (Sc^{III}), yet the corresponding **Ph{Ln}** can still be isolated (except for very labile **Ph{Sc}**)."

To summarize the answer to this question, there is a mismatch between the size of the metal ion and the ligand cavity, which results in poor inertness of the chelate, which, in turn, promotes side-reactions and complicates the analysis.

2. Can the authors comment on whether they see a possibility for Ln ion separation based on the kinetic differences shown in Fig 1?

Figure 1 shows the kinetics of ClickZip chelate formation. In our opinion, the differences between the Ln(III) ions in this property are insufficient to serve for any meaningful separation. Perhaps only for the extreme case of most distant La(III) and Lu(III) the much faster formation of the highly inert Lu(III) ClickZip chelate could serve to separate these two elements. However, this would not be

practical nor economical approach, as there are many methods that can achieve this separation more efficiently and conveniently.

The kinetics of acid-assisted dechelation (Supplementary Figs. 15-18) show slightly better prospects for possible separation. However, even here the separation of two adjacent lanthanides would be disappointingly incomplete and probably impractical. First, ClickZip chelates would have to be formed with the given mixture of lanthanides, which is a relatively slow reaction. Then, the formed chelates would have to be leached under specific acidic conditions, which for some of them is an extremely slow process. Overall, the whole procedure would probably be too slow compared to existing alternative methods of lanthanide separation (e.g. ion-exchange chromatography, extraction chromatography, etc.).

Reviewer #3 (Remarks to the Author):

The authors report an innovative approach to highly inert lanthanoid chelates, based on a complexation-triggered alkyne-azide cycloaddition. The intriguing and highly selective cycloaddition leads to the formation of cryptate-like Ln(III)-chelates, with an almost irreversible encapsulation of the metal ion.

The manuscript describes a monumental work with a careful and systematic investigation of the so-called ClickZip reaction with respect to the efficiency through the whole lanthanoid series (expanded to Sc and Y), the robustness towards the ligand substitution pattern and the isolation and characterization of the byproducts. The inertness of the Click-Zipped chelates is extensively studied and proved. The impressive stability of these chelates is demonstrated by their resistance to post-ClickZip chemical transformations and to the acidic hydrolytic conditions used for the complete hydrolysis of peptides. The last property has been exploited by tagging peptides with Ln(III)-chelates for in vivo studies, later recovering and quantifying the intact chelates.

In my opinion the manuscript deserves publication in Nature Communications, with only the following minor revisions.

Comments and corrections

1) All the metal chelates (along with the chelating agents L1s and their immediate macrocyclic precursors) have been isolated by reverse-phase preparative HPLC, even if the ClickZip reaction is demonstrated to be highly selective and efficient. This is a strong limitation to the potential applications of these useful highly inert chelates. I am well aware that macrocyclic chelates are really difficult to isolate and purify, sometimes requiring dedicated techniques, that are nevertheless been scaled up and industrialized. The authors should report at least an example of a larger scale preparation (1 mmol or more) of one of the L1chelating agents/metal chelates.

We acknowledge that HPLC purification may present a limitation for a potential scale-up of the synthesis to industrial quantities, though not an unsurmountable one. In our laboratory, we use preparative HPLC as a standard procedure for all products. This is to ensure that they are of high purity before being used for any further work. This is also the reason why we used the technique for preparation of the ClickZip chelates. Nevertheless, in most cases the ClickZip reactions proceed sufficiently cleanly that preparative HPLC purification is not necessary and could be replaced with a simpler and more scalable purification techniques.

As suggested here by the reviewer, we have conducted an additional experiment to scale up the ClickZip reaction to the requested >1 mmol amount. A short commentary was added to "Metal ion role and preferences" section and the results are detailed in Supplementary Figs. 6, 55 and 78. For this work we selected the chloro-derivative **CIL**¹ and Lu(III). This derivative was giving the lowest yields in the screening reported in Supplementary Fig. 3. As shown in the Supplementary Fig. 6, the reaction proceeded cleanly, providing only a single product. This is particularly evident when comparing the aromatic ¹H NMR signals of the reaction mixture (panel B) and the purified product (panel C). Overall, the product of the ClickZip reaction was obtained in excellent 91% yield and quantity of 1.19 mmol. We believe that these results demonstrate well the scalability of the ClickZip reaction.

2) Suppl. Fig. 3, Effect of L1 substitution on the Ln(III) ClickZip reaction. The ClickZip reaction of CIL1 with the early lanthanoids appears to be plagued by significant amounts of byproducts, with a clear difference of the reaction pattern observed in the ClickZip reaction of other L1s. Have the author an explanation for this difference? Have the authors any additional indication on the byproducts of this specific reaction?

As demonstrated in the previous answer, the ClickZip reaction is relatively fast and runs quite cleanly for the late lanthanides. However, its effectivity decreases towards the larger ions of early lanthanides due to a mismatch in size of the metal ion and the cage. As the overall reaction time increases for these metals, there is a higher chance for the alkyne and azide reactive groups of the open ligand and open chelate to engage in unwanted side reactions. This gives rise to many impurities, each with relatively low abundance. That said, the desired ClickZip chelate is the major product in all cases except Sc(III) ion.

This is generally true for all the explored L¹s, but as the reviewer correctly points out, the chloro-derivative CIL¹ has particularly low product/impurity ratio for the early lanthanides. We explain this by the electron-withdrawing character of the chloro-substituent, which makes the respective pyridine arm a worse donor, less strongly coordinating to the metal ion. As a result, the first step of the synthesis, which is a full coordination of the Ln(III) ion into the open ligand, is less efficient compared to the other L¹ derivatives. As a consequence, the whole ClickZip process is less efficient, takes longer time and that results in higher amount of impurities for the CIL¹ derivative.

Only the major impurities could be confidently identified by mass spectrometry and (if successfully isolated by preparative HPLC) also by ¹H NMR spectra. These are listed in Supplementary Fig. 2 and notably include the adduct with MOPS and a variety of dimeric species. Based on this information, one can assume that many of the remaining unidentified minor impurities are probably higher oligomers and/or combinations of the above (e.g. adduct of an open dimer with MOPS, etc.). The number of such combinations is very high. For the particular case of CIL¹, we have shown that the chloro-substituent can undergo surprising reduction with formate and to a small degree also nucleophilic substitution with hydroxide anion (both noted in Supplementary Fig. 23). Unfortunately, it was not possible to identify all the minor impurities.

3) The triazole ring in the structures of the 1,5-cz-[Ln(L1)] systems is drawn with a high degree of distortion. Please follow the IUPAC recommendations (10.1351/pac200880020277) for graphical representation standards for chemical structure diagrams, in order to have an aesthetic depiction. The same recommendations should be applied to the drawing of the neo-formed double bond in PhL1-MOPS (Suppl. Fig. 1, Suppl. Fig. 4) and to its corresponding (undefined) stereochemistry.

Regarding the MOPS adduct and its undefined stereochemistry, we have followed this reviewer's advice and updated the drawing accordingly (now in Supplementary Figs. 2 and 4).

Regarding the 1,5-cz-[Ln(L¹)] and related compounds, we would like to provide the reasons that led us to drawing the structures in this particular way, and the reasons why we believe it is important to preserve this way. These bicyclic molecules have roughly a spherical shape, which in any case requires drawing them with some perspective of depth to reflect the reality and make the drawing intuitively understandable. In principle, some parts of the molecule will always appear "distorted" due to their out-of-plane orientation. We have tried many versions of the drawing to satisfy the requirements of (i) easy reading and understanding, (ii) reflecting the real shape of the molecule as close as possible, and (iii) aesthetics. We found the current version as the best option. Crucially, it displays the two pyridine arms arranged symmetrically around the central metal ion and one of the triazole nitrogen atoms exactly above the metal ion, as they are in reality (this is best seen in Figure 3 and Supplementary Figure 20). When viewed from the side of the molecule, the pyridines are almost perfectly symmetrical, while the triazole ring is oriented nearly perpendicular to them. We believe that the symmetric arrangement of the pyridines is an important structural feature that contributes to the exceptional stability/inertness of the chelates and should therefore be preserved in the drawings.

However, this can only be achieved by drawing the triazole slightly distorted (as it truly is when seen from the side of the molecule). We believe that this necessary compromise is to no detriment of understanding - the single and double bonds making the triazole are still clear to understand and cannot be mistaken for any other structure. In contrast, drawing the triazole as a regular pentagon would require shifting the pyridines into an unsymmetrical arrangement not corresponding to the reality and also not pleasing aesthetically. For these reasons we wish to preserve the drawing as it is.

- 4) Page S30 and following pages, synthetic procedures,
- ¹H-NMR spectra. Please report coupling constants with one decimal digit.
- ¹³C-NMR signals should be better reported with the corresponding multiplicity (C/CH/CH₂/CH₃, referred to the non-decoupled ¹³C spectrum) rather than generic terms as “arom” and “mc”.

We have fixed all these issues.

- 5) Introduction: “in-vivo”, “in-vitro”, please remove hyphen

We corrected all cases to "*in vivo*", "*in vitro*" and "*ex vivo*".

- 6) Page 5: “proton(iz)ation”

Corrected to "protonation".

- 7) Page S12, the triazol(e) signal

Corrected to "triazole".

- 8) Page S30, desym(m)etrization

Corrected to "desymmetrization".

- 9) Page S30, please replace “TMS-≡” with the explicit form “ethynyltrimethylsilane” used afterwards.

Corrected to "ethynyltrimethylsilane".

- 10) Please use a uniform symbol for the volume (ml/mL) through the manuscript and the supplementary information.

We corrected all cases to "mL".

We thank all the reviewers for their suggestions and diligent work.

Point-by-point response to the REVIEWER COMMENTS

for second revision of the manuscript

"Ultra-inert lanthanide chelates as mass tags for multiplexed bioanalysis "

Reviewer #1 (Remarks to the Author):

The authors have provided detailed responses to all comments included in my previous report. They corrected a few minor issues and performed additional experiments to analyze the metal-assisted dissociation pathway. The results show that Zn(II) does not trigger complex dissociation at all, while Cu(II) does induce minor dissociation in very harsh conditions. Overall, these new results confirm the exceptional inertness of the complexes claimed in this paper. The manuscript is now ready for publication in Nat Commun.

We thank the Reviewer for the useful comments and suggestions.

Reviewer #2 (Remarks to the Author):

Thank you for addressing my comments

We thank the Reviewer for the useful comments and suggestions.

Reviewer #3 (Remarks to the Author):

Comments to revisions.

1) Preparation/RP-HPLC/larger scale.

The authors acknowledge that HPLC purification may present a limitation for a potential scale-up of the synthesis; personally, I agree that it is not an unsurmountable one with such an efficient transformation and favourable impurity profiles.

A larger-scale preparation is now reported, demonstrating as expected the scalability of the ClickZip reaction.

No issue to discuss.

2) Effect of L1 substitution on the Ln(III) ClickZip reaction.

The authors provide a reasonable explanation of the higher percentage of byproducts observed in the ClickZip reaction of CIL1 with early lanthanoids.

No issue to discuss.

3) Drawing, Triazole and MOPS-byproduct

I understand the difficulty, almost impossibility, to have a drawing of the whole molecule with the chelate moiety and the triazole both without distortions. I agree with the authors to give a priority to a correct drawing of the chelate substructure with a symmetrical arrangement of the pyridine rings, at the expense of the drawing of the triazole ring.

Thank you.

4) - ¹H-NMR spectra. Please report coupling constants with one decimal digit.

The authors reported the coupling constants with one decimal digit.

No issue to discuss.

Please specify the meaning of the non-standard abbreviation "dm" for the signals at 7.94 ppm and 8.05 ppm of compounds 12,13.

The abbreviation was corrected in both instances to “d”, meaning doublet.

5) ¹³C-NMR signals should be better reported with the corresponding multiplicity (C/CH/CH₂/CH₃) rather than generic terms as “arom” and “mc”.

The authors fixed most of the multiplicities. Some generic non-multiplicity terms (“Ph”) remain and should be corrected (see compounds: 6,7,8,9, PhL1, cz-1,4-PhL1, cz-1,5-PhL1, 1,4/1,4-cz-[Lu(PhL1)] and the diamagnetic Ph{Y}, Ph{Lu}, Ph{La}, Ph{Ca}, Ph{Lu-red}).

Thank you for catching these cases. We believe that we have now fixed all the multiplicity reporting accordingly.

6) All typos and other minor mistakes have been corrected.

No issue to discuss.

In my opinion the manuscript deserves publication in Nature Communications, once the remaining minor corrections are applied.

We thank the Reviewer for the useful suggestions and diligent work in finding errors.